# A Case Series of SARS-CoV-2 Reinfection in Elite Athletes

**DOI:** 10.3390/ijerph192113798

**Published:** 2022-10-24

**Authors:** Gábor Áron Fülöp, Bálint Lakatos, Mihály Ruppert, Attila Kovács, Vencel Juhász, Gábor Dér, András Tállay, Hajnalka Vágó, Boldizsár Kiss, Béla Merkely, Endre Zima

**Affiliations:** 1The Heart and Vascular Center, Semmelweis University, H-1122 Budapest, Hungary; 2Department of Sports Medicine, Semmelweis University, H-1122 Budapest, Hungary

**Keywords:** COVID-19, SARS-CoV-2, athletes, reinfection

## Abstract

Objectives: The actual frequency and the risk factors of SARS-CoV-2 reinfection is still a matter of intense scientific discussion. In this case series, we report three elite athletes who underwent COVID-19 reinfection with a short time frame. Case presentations: As a part of contact tracing, three speed skaters (22-, 24-, and 29-year-old males) were found to be SARS-CoV-2 positive by polymerase chain reaction (PCR) tests. Later on, only one of the athletes experienced mild symptoms, such as fatigue, loss of smell and taste and subfebrility, while the other two athletes were asymptomatic. Following the quarantine period, detailed return-to-play examinations, including laboratory testing, ECG, 24-h Holter monitoring, transthoracic echocardiography and cardiac magnetic resonance imaging, revealed no apparent abnormality; therefore, the athletes restarted training. After a median of 74 days, all three athletes presented with typical symptoms of COVID-19, such as fever, marked fatigue and headache. SARS-CoV-2 PCR tests were performed again, showing recurrent positivity. Repeated return-to-play assessments were initiated, finding no relevant abnormality. Athletes were also tested for SARS-CoV-2 anti-nucleoprotein antibody titers, showing only modest increases following the second infection. Conclusions: We report a small cluster of elite athletes who underwent a PCR-proven SARS-CoV-2 reinfection. According to these findings, athletes may be considered as a high-risk group in terms of recurrent COVID-19.

## 1. Introduction

The SARS-CoV-2 pandemic is an unprecedented challenge for healthcare systems worldwide, requiring many novel approaches from individual patient management to the level of population medicine, e.g., isolation policy and contact tracing. Nowadays, the carefully titrated social-distancing measures, novel treatment options and, most importantly, vaccination programs are the established methods of primary and secondary prevention. Still, important counteracting factors also must be taken into account, such as the appearance of novel mutations and the possibility of reinfection [1,2,3]. The importance of the former one is robustly underpinned by the events of the recent months; however, data are still scarce discussing the phenomenon of reinfection: mostly isolated cases are reported in the literature; therefore, little to no information is available regarding its determining factors and the actual frequency. Moreover, large population studies are also controversial in terms of the “typical” time frame between the two infections [2,3].

SARS-CoV-2 reinfection is a highly relevant issue for elite athletes as well, regardless of their vaccination status. There are several uncertainties regarding COVID-19 infection in this distinct population, such as the negative effects of the disease on their overall health [4] and finding the adequate time for return to play [5]. The psychological and economic burdens of the cancelled competitions are also far from negligible [6]. Evidence suggests that the rate of upper respiratory tract involvement is more frequent in athletes compared to the general population, potentially increasing the risk of infection and also reinfection with SARS-CoV-2 [7]. The latter has a consequence of an obligatory recurrent detraining and loss of efficacy and strength.

In this paper, we report a case series about short-term (less than 2 months) reinfection of three elite athletes from the Hungarian national short-track speed-skating team, and, additionally, we present their return-to-play examination results after each infection, focusing on the effects of reinfection on their health status.

## 2. Case Presentations

Two athletes (Athlete 1: 29-year-old male; Athlete 2: 24-year-old male) tested positive for SARS-CoV-2 with rt-PCR (HBRT-COVID-19; Chaozhou Hybribio Biochemistry Ltd.; Chaozhou, China) on 9 September 2020. The tests were performed as a part of contact tracing. After the positive results of Athletes 1 and 2, an additional contact tracing was indicated: as a result, Athlete 3 (22-year-old male) tested positive for SARS-CoV-2 with rt-PCR on 15 September 2020. Overall, two of the athletes showed no symptoms, while Athlete 1 was subfebrile for two days, mentioned fatigue lasting three days and also presented with loss of smell and taste. All of them were sent to self-isolation. At the time of their infection and reinfection, the athletes were not vaccinated against COVID-19 and, since they had mild symptoms, no specific treatment was applied while they were isolated.

Twenty days following the first positive rt-PCR of Athletes 1 and 2, and 14 days after the infection of Athlete 3, none of the subjects mentioned any complaints; therefore, return-to-play examinations were initiated. Resting 12-lead ECGs did not reveal any pathological abnormalities or apparent changes when directly compared to previous data. Twenty-four-hour Holter data (LabTech Cardiospy 5.04.01., LabTech Ltd., Debrecen, Hungary) showed normal circadian rhythms, with a lower normal heart rate because of athletic adaptation. Minimum and maximum heart rate values did not remarkably differ from normal ranges; besides, no conduction or impulse generation disturbances were observed. Notable changes in the QT interval range and the heart rate variability were not present. Either no or a clinically non-relevant percentage of premature atrial or ventricular complexes, or malignant arrhythmias, were recorded.

Laboratory tests also showed no relevant alterations; normal inflammatory marker levels were seen along with normal hs-Troponin T and D-dimer levels (Figure 1). Antibody measurement was not performed, since no robustly validated antibody test was available at the time of the examinations. 3D Transthoracic echocardiography (GE Vivid E95 ultrasound system, 4Vc-D transducer; GE Healthcare, Horten, Norway) showed maintained left and right ventricular function with no regional wall motion abnormality, normal estimated pulmonary systolic pressures, and the absence of valvular disease. Cardiac MRI was also performed (Siemens MAGNETOM Area 1.5T; Siemens Healthineers, Erlangen, Germany), confirming maintained left and right ventricular global function with no regional wall motion abnormality. The presence of edema or necrosis/fibrosis was not supported by T1/T2 mapping or late gadolinium enhancement, ruling out myocarditis or pericarditis in the three athletes. Compared to the previous assessments, mildly decreased wall thicknesses were seen, which correspond to the effects of detraining in parallel with the suspended athletic activity during infection. In accordance with the findings, the athletes were allowed to return to training.

After 74 and 80 days following the positive rt-PCR tests, respectively, all three athletes presented with typical symptoms such as fever, marked fatigue, and headache following contact with a COVID-19-positive coach. On 23 November 2020, they were tested positive again for SARS-CoV-2 with rt-PCR. Opposite to the first infection period, they showed more intense and prolonged symptoms; however, hospitalization was not necessary in any cases. Symptoms lasted for an average of 6.5 days. Repeated return-to-play assessments were initiated following the reinfections as well. Ten days following the second infection, the athletes had fully recovered from their symptoms. We have repeated the examinations showing no apparent abnormalities. ECG and Holter results were unremarkable, while echocardiographic and cardiac MRI studies also revealed practically identical morphological and functional measures to the previous return-to-play assessments. Of note, we found mildly elevated D-dimer (0.68 ng/mL) and elevated ferritin values (323 µg/L) in Athlete 1, which may correspond to the typical consequences of COVID-19 (Figure 1). Nevertheless, as Athlete 1 was lacking any remaining symptoms, no specific treatment was considered to be necessary.

Since the quantitative measurement of serum SARS-CoV-2 IgG antibodies have been already made available as a part of the second return-to-play assessments, all of the athletes underwent IgG measurements as well (Architect i2000SR, Abbott Laboratories, Irving, TX, USA), revealing modest positivity in terms of the antibody levels (Figure 2).

## 3. Discussion

While the exact mechanisms or even the actual possibility of SARS-CoV-2 reinfection is still a matter of debate, evidence is growing that the immune response following COVID-19 may not exclude the probability of reinfection or reactivation [2,3]. Still, due to the low number of documented cases, the data are scarce about which populations may be more prone to reinfection. Consequently, there is very limited information about the possible outcomes of a second COVID-19 infection. In our case series, we reported three professional speed skaters with recurrent SARS-CoV-2 rt-PCR positivity and consequent symptoms along with no significant cardiopulmonary sequaleae.

Several studies aimed to assess the immunology of COVID-19. According to our current knowledge, the vast majority of PCR-positive individuals (approx. 91–99%) seroconvert, and the antibody titers usually persist for at least 6–8 months [8]. It is important to mention that beyond the humoral response, cellular immunity also plays an important role in protection with a comparable longevity [9]. On the other hand, it is also suggested that as antibody titers are decreasing over time, the possibility of a reinfection may increase in parallel [10]. We must underline that the majority of the reported reinfection cases were diagnosed at a minimum of three months after the first infection. Recently published momentous studies examining nationwide healthcare databases suggest that the SARS-CoV-2 reinfection is a rare phenomenon; moreover, compared to our case series, they mostly report a longer time frame between the infections [2,3]. On the other hand, there are data that support the possibility of prolonged viral shedding and COVID-19 positivity in a short time frame between the infections, although no genome sequencing was performed to support the data [11]. Long-term viral shedding is mostly common in severely immunodeficient individuals, just like post-transplanted patients [12].

Still, there are multiple factors that may explain the early reinfection in our cases. First, asymptomatic COVID-19 infection is usually associated with lower antibody titers, and the titer seems to correlate with the protection [13]. As our athletes were demonstrating no or just very mild symptoms (i.e., fatigue and loss of smell and taste) at the time of the first COVID-19 infection, they may have a less pronounced and/or more rapidly declining adaptive immune system response. Moreover, the antibody titers following the second infection were also showing only modest increases in our cohort (Figure 2), as the circulating antibody levels were markedly lower in all of our subjects compared to the typically observed values [14]. Interestingly, as one can appreciate in Figure 2, one of our athletes showed no antibody titer increase even after the second infection, which might be due to the lack of major symptoms or smaller viral load. Importantly, non-human primate studies suggest that SARS-CoV-2 does not generate sterilizing immunity: despite the markedly reduced viral load, in monkeys that underwent COVID-19, viral RNA, as a marker of active replication, was detectable from the respiratory tract following a reinfection protocol [15]. Second, regular physical exercise is associated with significant changes of the immune regulation as well. Traditionally, it was thought that intense physical training and especially overtraining provoke suppressed immune reaction; however, recent studies rejected this theory [16]. Still, physical exercise has marked immunomodulatory effects, which may increase the risk of SARS-CoV-2 reinfection [7,16]. Noteworthy, habitual exercise at an intense level might suppress the activity of the mucosal immunity and also result in decreased secretory and circulating IgA levels [17]. This is of particular interest, since the mucosal immune system has been lately recognized as a crucial defensive barrier against COVID-19 infection [18].

To address the limitations of the study, we must mention that no genome sequencing was performed on the nasopharyngeal swab samples; therefore, we cannot exclude the possibility of short-term in vivo evolution of the virus strain. On the other hand, with more than two months after the first infection, the chance of this phenomenon is fairly low, especially in three parallel cases. We also have to consider the potential false positivity of the rt-PCR tests, a phenomenon that is markedly neglected despite the extremely large number of rt-PCR assessments worldwide [19]. Nonetheless, our athlete cohort trained together regularly, demonstrated symptoms and also positive rt-PCR at the same time during the first and also the second infection, which is highly suggestive of an actual spreading of SARS-CoV-2 in a small community with consequential true positive tests. Moreover, robust evidence demonstrates that a positive rt-PCR test confirms COVID-19 infection with an excellent positive predictive value [20]. A further limitation might be the development of COVID-19 vaccines, which markedly decreased the chances of infection, reinfection, or severe disease progression. The athletes reported in this manuscript were vaccinated in March 2021. With COVID-19 vaccines, such a short-term reinfection might be avoided. Second, the appearance and spread of new COVID-19 variants might affect the chances of infection and reinfection. At the time of the report, the Alpha (B.1.1.7) variant was prominent in Europe, while in 2022 mostly Omicron (B.1.1.529) variants are present in Europe. On the other hand, Omicron variants are believed to be more infectious than previous variants, which might increase the chance of reinfection even after vaccination [21].

The return-to-play examinations revealed no medically relevant alterations associated with COVID-19 and the athletes did not have persistent complaints and could continue training. It is particularly important in the face of the worrisome data, showing that long-lasting aspecific symptoms are fairly common, even in young individuals, following an uncomplicated infection [22]. Initial reports also demonstrated a high frequency of SARS-CoV-2-associated myocarditis in athletes [23]; however, further data did not support this finding [24]. According to our knowledge, there are no reports of early COVID-19 reinfections in elite athletes; on the other hand, there is quite a large amount of data about SARS-CoV-2 positivity in athletes and its possible healthcare consequences. For example, data from France suggest that most of the asymptomatic or mildly symptomatic COVID-19 infections have no cardiac sequelae in athletes; thus, a need for a thorough cardiac examination protocol as part of the return to play might be questionable [25]. Another study on almost 800 athletes also showed a low probability of cardiac involvement because of COVID-19 infection [4]. As more and more data are pointing i the same direction, guidelines on a return-to-play protocol tend to be less strict in terms of cardiovascular examination of the athletes [26]. On the other hand, some authors found that COVID-19 infection might lead to mild vascular impairment, affecting athletic performance [27]. As there is a lack of data about short-term reinfection in the population of elite athletes, we believe that should not discard the importance of a return to play examination protocol. 

## 4. Conclusions

In our case series of three professional speed skaters, elite athletes underwent SARS-CoV-2 reinfection with more severe, albeit far from life-threatening symptoms, at the time of the second rt-PCR positivity. Compared to the majority of the documented COVID-19 reinfection cases, the second infection occurred within a relatively short time frame, only 2 months following the initial detection of the disease. Importantly, we were the first to report that reinfection had no further deleterious cardiovascular sequela. In conclusion, elite athletes may be a high-risk population in terms of a SARS-CoV-2 reinfection rate because of extreme training conditions and should be assessed more thoroughly; however, large-scale data are definitely needed to confirm this theory.

### 4.1. Practical Applications

This report draws attention to the possibility that elite athletes might be more prone to COVID-19 reinfection. Data are still scarce about short-term COVID-19 reinfection in elite athletes, and the possible health-related consequences are not well-characterized. As more and more data became available in athletes after COVID-19 infection, the return-to-play protocols became less strict and thorough, allowing athletes to start training early after COVID-19 positivity in cases of mild or no symptoms [26]. On the other hand, there is still a lack of data regarding the effects of short-term reinfection, which might need a more detailed return-to-play examination protocol. Our data also underlines the importance of regular testing, even in the era of mildly symptomatic or asymptomatic COVID-19 infection.

### 4.2. Possible Future Directions

Even though we found no major pathological alteration of the three athletes reported in this manuscript, we still believe that a more detailed return-to-play protocol should be performed in such cases. Reports like ours are crucial to understand the effects of such a disease course in elite athletes, especially in the times of new COVID-19 variants.

## Figures and Tables

**Figure 1 ijerph-19-13798-f001:**
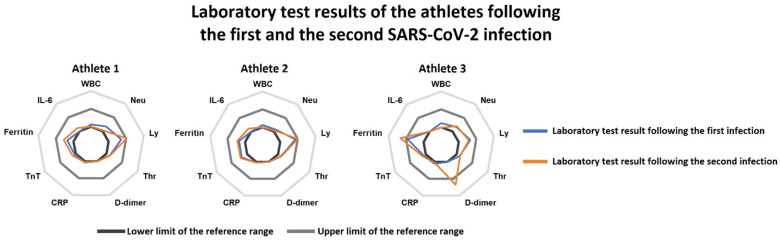
Radar charts of the laboratory tests following the first SARS-CoV-2 infection (blue line) and the reinfection (orange line) in the three athletes. Athletes 2 and 3 had practically no relevant alterations at the first and the second examinations. Athlete 1 presented with increased serum ferritin levels at the first examination, which persisted at the second return-to-play assessments as well. Moreover, mildly elevated D-dimer levels were also found at the second laboratory test in this subject. Abbreviations: WBC: white blood cell count, Neu: neutrophil count, Ly: lymphocyte count, Thr: thrombocyte count, CRP: C-reactive protein, TnT: troponin-T, IL-6: interleukin-6.

**Figure 2 ijerph-19-13798-f002:**
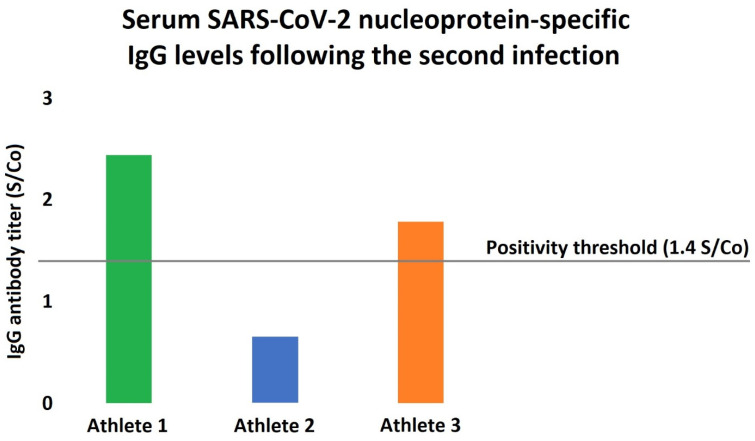
Serum SARS-CoV-2 nucleoprotein-specific antibody levels in the athletes following the reinfection. Humoral immunity was assessed following the second infection, showing only modest increases in the antibody levels. Interestingly, Athlete 2 had an antibody titer significantly below the positivity threshold even after the reinfection, while Athlete 3 also barely exceeded the accepted antibody-positivity level.

## Data Availability

All the data are based on medical records of the athletes, which we gladly provide upon request. Requests should be sent to the corresponding author.

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
