# Peer review of "A Case Series of SARS-CoV-2 Reinfection in Elite Athletes"

_ijerph, 2022, doi:10.3390/ijerph192113798_

Round 1
Reviewer 1 Report
Dear Authors.
The following is a review of the article entitled "A case series of SARS-CoV-2 reinfection in elite athletes". In this paper, we report a case series about short term (less than 2 months) reinfection of three elite athletes from the Hungarian national short track speed skating team, and additionally, we present their return to play examination results after each infection, focusing on the effects of reinfection on their health status. Thank you very much for thinking of me as a reviewer for this study.
After carefully reading the manuscript, I have the following comments and suggestions for the authors: I find the manuscript interesting and novel. The theoretical framework is correct. The methodological development is correct. Correct presentation of the results.
Add conclusions, practical applications, and limitations. It is suggested to add possible future perspectives.
The discussion should focus on comparing the results obtained with the results of other studies in young athletes.
I look forward to the publication of your work.
Regards
Author Response
First of all, we would like to express our gratitude for the comments, as by answering them, we could improve the quality of the manuscript.
We extended the limitations part of the manuscript, as well as added a practical applications and a future perspective paragraph and extended the discussion paragraph with further data focusing on data obtained from other studies on the topic, and aimed to compare our results with those findings.
We hope, that we could address all the comments of the reviewer but if needed we are happy to answer further comments.

Reviewer 2 Report
This is a well written manuscript that provides interesting information about early COVID-19 reinfection in elite athletes and performs analyses for relevant factors. Some revisions are suggested as follows:
- Please clarify that the short-term recurrent SARS-CoV-2 PCR positivity is a consequence of true reinfection, rather than prolonged viral shedding (Xiao et al., 2021) or other conditions such as suppressed but not fully cleared virus after the first infection.
- More information on the case series needs to be provided. For instance, did the athletes get vaccination before the infection/reinfection? Were there any medical treatments received during the self-isolation periods?
- Line 14. “29-year-old”
- Lines 23-24. Weren’t antibody measurements only available after the second infection?
- Line 42. The sentence needs to be rephrased, since the Tokyo Olympic Games were held a year ago.
- Line 191: Could the authors explain how to analyze the reinfection rate given that only three athletes were reported in the current study?
- Figure 1. The quality of the figure needs to be improved - the current version is not very clear.
- Figure 2: It is interesting to see that Athlete 2 had an antibody titer significantly below the positivity threshold. Could the authors provide possible reasons for this?
Reference
Xiao, C. H., Chen, L. F., & Li, Y. (2021). Recurrent SARS-CoV-2 RNA positivity and prolonged viral shedding in a patient with COVID-19: a case report. BMC infectious diseases, 21(1), 1-6.
Author Response
First of all, we would like to express our gratitude for the comments, as by answering them, we could improve the quality of the manuscript.
- Please clarify that the short-term recurrent SARS-CoV-2 PCR positivity is a consequence of true reinfection, rather than prolonged viral shedding (Xiao et al., 2021) or other conditions such as suppressed but not fully cleared virus after the first infection.
This comment is especially important, as to differentiate between early reinfection and prolonged SARS-CoV-2 PCR positivity and possible reactivation of the virus is crucial. As we mention in the limitations part, that no genome sequencing was performed on the nasopharyngeal swab samples, therefore, we cannot exclude the possibility of short-term in vivo evolution of the virus strain, on the other hand, with more than two months after the first infection the chance of this phenomenon is fairly low, especially in three parallel cases. While Xiao et al. propose the possibility of short term COVID-19 positivity due to viral shedding, the authors also did not perform genome sequencing thus the possiblity of reinfection in that case can not be excluded either. While we must not exclude the possibility of a prolonged viral shedding, it is mostly present in immunodeficient individuals, such as post-transplanted patients. While elite athletes who undergo regular heavy training might develop vulnerability against upper respiratory tract infections, such a prominent immunodeficient effect as it is observed in the post-transplanted patients due to their regular medications, is unlikely. Nonetheless, our athlete cohort trained together regularly, demonstrated symptoms and also positive rt-PCR at the same time during the first and also the second infection, which is highly suggestive of an actual spreading of SARS-CoV-2 in a small community with consequential true positive tests.
In conclusion we can say, that although no genome sequencing was performed in order to prove the reinfection, the possibility of prolonged viral shedding is fairly low in our cohort.
- More information on the case series needs to be provided. For instance, did the athletes get vaccination before the infection/reinfection? Were there any medical treatments received during the self-isolation periods?
It is again a very important comment, as vaccinations and tretment such as remdesivir or monoclonal antibodies might change the course of the disease. At the time of the infection of our athlete cohort, the COVID-19 vaccines were not availabe in Hungary and since their infection was asymptomatic/mildly symptomatic we did not apply any specific medication during their isolation period. We updated the manuscript with this additional information.
- Line 14. “29-year-old”
We corrected the text.
- Lines 23-24. Weren’t antibody measurements only available after the second infection?
We corrected the text.
- Line 191: Could the authors explain how to analyze the reinfection rate given that only three athletes were reported in the current study?
This statement was based on the cited manuscript and not based solely on our data reported in the manuscript.
- Figure 1. The quality of the figure needs to be improved - the current version is not very clear.
We updated the figure.
- Figure 2: It is interesting to see that Athlete 2 had an antibody titer significantly below the positivity threshold. Could the authors provide possible reasons for this?
The final comment of the reviewer is also very important, as the lack antibody response might be a reason for increased vulnerability against the virus. The lack of antibody response might be due to the mildly smptomatic disease course. On the other hand, the other althletes had mild symptoms as well, but still developed higher antibody response. This might be due to the difference in viral load, which is hard to assess. A possible indicator for viral load might be the CT number of the nasopharyngeal sample collected from an individual, although the CT number is highly dependent on the sample collection. As a result, we can only speculate that the possible reason for the lower antibody titers might be also due to the difference in viral load. We updated the manuscript with this additional information.
We hope that we were able to address the concerns of the reviewer adequately. We are delighted to address any further comments.

Round 2
Reviewer 2 Report
The authors have addressed my concerns and I have no more comments.